# Influence of the Fly Ash Content on the Fresh and Hardened Properties of Alkali-Activated Slag Pastes with Admixtures

**DOI:** 10.3390/ma15030992

**Published:** 2022-01-27

**Authors:** María Jimena de Hita, María Criado

**Affiliations:** Instituto de Ciencias de la Construcción Eduardo Torroja, Spanish National Research Council (CSIC), Serrano Galvache 4, 28033 Madrid, Spain; mariajimena.dehita@ietcc.csic.es

**Keywords:** alkali-activated slag/fly ash, admixtures, workability, setting time, stability, mechanical strengths

## Abstract

A study on the influence of the inclusion of slag or fly ash and five types of superplasticizers on the fresh and hardened properties of alkali-activated cements is presented. Three alkali-activated slag formulations with different fly ash content (0, 15, and 30%) in the presence of five admixtures (vinyl copolymer, melamine, and three polycarboxylates with different chain lengths) were assessed for fluidity control and setting adjustment without loss of mechanical properties. Solid sodium metasilicate was used as an alkaline activator. Their fresh and hardened properties were studied through slump, setting time, isothermal calorimetry, mechanical strengths, and porosity tests. The results showed that the increase of fly ash content delayed the reaction and improved workability but reduced compressive strengths. Concerning the admixtures, these maintained fluidity especially for the one based on polycarboxylate with very long chains. The melamine and polycarboxylate with very long chain admixtures did not have a drastic impact on mechanical properties at early ages; even a gain of flexural and compressive strength was noted.

## 1. Introduction

Alkali-activated materials can be produced by one of two procedures: the one-part mix system and the two-part mix system. In the one-part or “just add water” system, which is new, both the aluminosilicate precursor and alkaline activator are in their solid state [1,2], which requires the addition of water to act as a cementitious material. In the two-part system, the most commonly used to date, a solid precursor is mixed with a liquid alkaline activator [3,4,5]. Several problems associated with the liquid alkaline activators have been observed as they are highly corrosive and viscous, making transportation and handling difficult in their operation.

In the past few decades, the production of alkaline-activated blast furnace slag and class F fly ash has been extensively studied. Both these alkali-activated cementitious materials present a series of advantages and disadvantages according to different chemical reactions and curing conditions. Among the main characteristics of alkali-activated slag are high strength, rapid setting, impermeability, and improved fire resistance [6,7,8], while high-strength, high-heat and acid resistance [9,10,11,12,13] are alkali-activated fly ash features.

However, fly ash requires heat treatment to be activated and set due to its low activity at room temperature. This problem can be overcome by adding slag, boosting the reactivity of the system, and increasing the strength due to the coexistence of alkaline aluminosilicate hydrated gels, usually N–A–S–H with C–A–S–H [14,15]. In addition, a higher slag content as a replacement for the fly ash in the system also favors matrix densification, resulting in a reduced chloride migration rate in concrete [16]. In the case of the alkali-activated slag, the type and the chemical constitution of the alkaline activator strongly affect the reaction kinetics and strength development, leading to workability loss and rapid setting [17]. The incorporation of fly ash with a lower percentage of CaO lengthens the setting time and prevents rapid hardening [18,19].

If these alkali-activated materials are considered as binders for most construction operations [20,21,22,23,24], a selection of chemical admixtures is the best option for fluidity control and setting adjustment without losing mechanical properties. These admixtures are designed for Portland cement and plasticize and fluidize the concrete through an electrostatic or steric mechanism that causes repellency between cement particles [25]. However, their effectiveness in alkali-activated cement is not assured due to the pore solution of these cements with a higher pH and ionic strength, which poses a prejudice to the stability of admixtures. The first concern is physical stability related to the agglomeration, color change or separation into layers of superplasticizers in the alkaline activator [26]. The second one is chemical stability because the molecular structure of the admixtures changes in highly alkaline activator solutions [26]. In previous studies [27,28], chemical structural changes were observed in some superplasticizers when the pH of the alkaline medium was up to 13. Under these conditions, esters and amides underwent alkaline hydrolysis, while the ethers and aromatics were stable.

Few studies have been developed to investigate the properties of fresh alkali-activated slag/fly ash pastes in the presence of chemical admixtures [2,29,30,31,32,33,34]. If a fixed percentage of slag and fly ash is taken into account, the results obtained by Keulen et al. [29] showed that significantly improved workability and strength development of fly ash/slag (73.7/25) concrete were obtained by an increased polycarboxylate admixture content. Alrefaei et al. [30] evaluated the effectiveness of three different types of superplasticizers in a one-part alkali-activated slag/fly ash (50/50) material using anhydrous Na_2_SiO_3_ as the activator. All of them significantly improved the flowability and marginally affected the compressive strength. For a high water/precursor (≥0.36), polycarboxylate was effective, while naphthalene performed better as a low water/precursor (≤0.36). Oderji et al. [2] focused their work on the investigation of a one-part fly ash/slag (85/15) binder cured at room temperature to obtain the best workability with reasonable mechanical strength in the presence of 2 wt.% of borax, sodium triphosphate, polycarboxylate, sodium gluconate, sodium lignosulphonate, and calcium lignosulphonate. The results revealed that sodium gluconate had lower mechanical strength compared to borax but exhibited comparable workability. Sodium triphosphate had lower workability to borax but exhibited comparable mechanical strength. In addition, both lignosulphonate-based admixtures were quite ineffective since they led to a marked decline in mechanical properties and did not improve workability. Finally, the polycarboxylate had a behavior similar to that of the control mix without admixture.

On the other hand, there are other studies that consider the effect of inclusion of different contents of slag or fly ash, Jang et al. [31] reported that polycarboxylate shows a retarding effect on alkali-activated fly ash/slag pastes (0/100, 30/70, 50/50, 70/30, and 100/0) with negligible effect on the heat of hydration, and improves workability more significantly than naphthalene. A content of more than 2 wt.% polycarboxylate positively affected mechanical development before 7 days. Laskar et al. [32] observed that the addition of fly ash (0, 20, 30, 40, and 50%) contributed to achieving satisfactory compressive, workability and bond strength of the geopolymer concrete up to a certain percentage of substitution. Polycarboxylate showed better workability, compressive and bond strength than naphthalene at all dosage levels. A study to examine the effect of dosage of superplasticizers (0 to 3%) on the rheological properties of alkali-activated fly ash and slag (0/100, 25/50, 50/50, 75/25, 100/0) composites with varying alkali/binder ratios was conducted by Namitha et al. [33]. The setting time increased marginally with the dosage of polycarboxylate. The use of naphthalene with an increase in a/b ratio was not very effective in prolonging the setting time of alkali-activated pastes. Moreover, Raju et al. [34] investigated the effect of the inclusion of slag and the type and dosage of superplasticizers on the fresh properties of alkali-activated fly ash paste (0, 25, 50, 75, 100). An increase in slag content and a decrease in alkali content reduced setting time and workability. Naphthalene performed better than polycarboxylate in the mini-slump test for all the mixes. In the case of the setting time test, naphthalene showed more effectiveness at retarding the set for the 100% slag paste.

This study was carried out within the framework of the IRINEMA project (Immobilization of Nuclear grade Ion Resins in Alkali-activated Materials, 2019-T1/AMB-13672). The aim of it was to find an alkali-activated material to replace the Portland-based cement currently used to immobilize nuclear waste. Slag and ash proportions were chosen as a starting point for the formulation currently used in Spain for this purpose, which is approximately 70% OPC–30% FA. It was decided to substitute the OPC for BFS to carry out comparative studies (70% BFS) and to evaluate the behaviour of a total substitution by slag (100% BFS) and an intermediate value to evaluate possible trends (85% BFS).

To avoid the flash-setting problem and improve fluidity, three alkali-activated slag formulations with different fly ash content, together with the presence of five admixtures (vinyl copolymer, melamine, and three polycarboxylates with different chain lengths), were developed. Their fresh and hardened properties were studied through slump, setting time, isothermal calorimetry, mechanical strength, and porosity tests. To carry out this study, new commercial admixtures were evaluated: a vinyl copolymer with ester groups in its structure and a polycarboxylate, PC3, the formation chemistry and structure of which are different from traditional polycarboxylate ether admixtures. Since they could perform better than commercial admixtures designed for Portland cement in highly alkaline media and solve their denaturation problem. Moreover, the addition of admixtures was carried out in two stages to prevent these chemical structural changes and maintain fluidifying properties.

## 2. Materials and Methods

### 2.1. Materials

In this study, Spanish blast-furnace slag (BFS, Calumite Ibérica) and fly ash from a Spanish power plant (FA) were waste material used as binder material. The as-received slag was milled in a mechanical ball mill so that 96% of the particles would be smaller than 45 µm to avoid coarse granules and increase reactivity while conducting the tests. The chemical composition of two raw materials was determined by X-ray fluorescence spectroscopy (XRF, S8 Tiger Bruker), see Table 1. The specific surface area by laser granulometry (MASTERSIZER S) and average particle sizes by BET method [35] (ASAP 2010, Micrometrics Instrument Corporation, Norcross, GA, USA) of the slag and fly ash were 1154 and 2023 m^2^/kg and 3.8 and 38.4 µm, respectively.

The alkaline solution was sodium metasilicate powder (Na_2_SiO_3_, Sigma-Aldrich, Barcelona, Spain) with a concentration of 7 wt.% by mass of binder [36]. Sodium silicate was chosen as an alkaline activator in this work because previous studies [37,38,39] reported that its use lead to the formation of alkali-activated material with good mechanical and durability performance. This activator is solid and the solution can be produced in situ to avoid transportation problems in the operation. The liquid-to-solid (anhydrous slag/fly ash + anhydrous sodium metasilicate) ratio was 0.45.

The admixtures that were used in this study were vinyl copolymer (V), melamine (M) and three polycarboxylates (PC1, PC2 and PC3) of different chain lengths. All of them were supplied by Sika Spain. The characteristics and dosage of the admixtures are shown in Table 2.

### 2.2. Preparation of the Alkali-Activated Pastes and Mortars

Three formulations of blend pastes and mortars were made with slag/fly ash in the ratio of 100/0 (referred to as 100% BFS), 85/15 (referred to as 85% BFS) and 70/30 (referred to as 70% BFS) by mass and then activated with sodium metasilicate. To ensure the homogeneity of these mixtures, they were introduced into a turbula for 1 h, and their activation was carried out in two stages. Prior to the activation process, the admixture was pre-dissolved in half the water to avoid denaturation when it came in contact with the activator, whereas the activator was dissolved in the other half of the water under magnetic stirring. In the first stage, the dissolved admixture was incorporated into the blend and the paste was mixed for 2 min at 500 rpm. In the second one, the activator solution was added and mixed for another 2 min at 500 rpm. To produce the mortar specimens (4 × 4 × 16 cm^3^), the sand/binder ratio was held at 3. A standardized, evenly graded siliceous sand was employed (SiO_2_ content of 99%). Mortars were sealed with cling film and cured at 20 ± 2 °C for 2 and 28 days, after which they were subjected to different tests.

### 2.3. Testing Procedure

Several tests were conducted to determine the fresh properties of the alkali-activated slag/fly ash pastes and analyse the admixture stability in the alkaline solution.

Mini-slump tests consisted of measuring the expansion (diameters) of materials on a flat methylmethacrylate plate at zero time, the time of their mixing, and at 30 min intervals for 2 h in this study. The paste was introduced into a truncated conical mould (19 × 37.5 × 57.5 cm^3^) and after one minute, the mould was lifted in the vertical direction and the diameter of the spread pat was measured using a caliper [40].

The setting time test consists of the periodic introduction of a standard needle into a cementitious material, and after analysing its specific resistance to penetration, the initial and final times of the setting period were determined according to European Standard UNE-EN 196-3 [41]. An automatic Vicat Needle (AUTO-VICAT, Ibertest, Madrid, Spain) was employed, and the tests were carried out on the same formulations that were used in the mini-slump tests.

Stability tests were conducted on the five admixtures in the sodium metasilicate solution (pH = 13.4) using Fourier transform infrared spectroscopy (FTIR, Thermo Scientific Nicolet 6700, Waltham, MA, USA), to ascertain how their chemical structures had been affected by a highly alkaline medium. The alkaline solution-to-admixture ratio was held at 1. The mixtures were placed in an oven at 120 °C for 24 h to remove excess water and then put into a plastic bottle at room temperature for 24 h to achieve total drying. KBr pellets (200 mg of KBr and 0.5 mg of admixture) were used to record the spectra. Spectral analysis was performed over a 4000–400 cm^−1^ range at a resolution of 4 cm^−1^ (64 scans).

Subsequently, the two most effective admixtures were selected to continue with a more in-depth study of the fresh state through isothermal calorimetry and the solid state through mechanical strengths and porosity.

The evolution of reaction kinetics was monitored using a TAM Air isothermal calorimeter at 25 °C. The fresh pastes were performed externally and manually (2 min) and then 5 g of each paste was transferred immediately into the calorimeter ampoule to record the heat flow. The experiments were conducted during the first 5 days after mixing and all values of the heat release rate were normalised by the total mass of paste analysed.

Mechanical strengths, flexural and compressive, at 2 and 28 days were determined using an Ibertest (Autotest-200/10—SW) testing frame according to European standard UNE-EN 196-1 [42].

The pore size distribution and pore volume of the mortars at 28 days were analyzed using Micrometrics Autopore IV 9500 porosimeter. The pore diameter was derived using Washburn’s law: D = (−4cosθ)γ/P, where D is the pore diameter (µm); θ is the contact angle between the fluid and the pore mouth (141.3°); γ the surface tension of the fluid (485 N/m); and P is the applied pressure to fill the pore with mercury (MPa). The maximum pressure applied was 227 MPa.

## 3. Results and Discussion

### 3.1. Influence of the Fly Ash Content on the Fresh Properties of Alkali-Activated Pastes with Admixtures

Initially, slump and setting time tests were conducted to determine the fresh properties of the alkali-activated slag/fly ash pastes and the influence exerted by the fly ash content and type of admixtures (vinyl copolymer (V), melamine (M) and three polycarboxylates (PC1, PC2 and PC3) in these parameters. Subsequently, a stability study of the admixtures was carried out to explain the behaviour developed by them in this highly alkaline medium.

#### 3.1.1. Setting Time

Figure 1 shows the variation in initial and final setting times of alkali-activated slag/fly ash mixtures with variation in fly ash content and chemical admixture. The setting time of the 100% BFS sample was an initial time of 9.8 h and a final time of 11.8 h, while those of the other samples were delayed: 10 h and 13.7 h for 85% BFS and 19 h and 27.3 h for 70% BFS. A higher proportion of fly ash in the alkali-activated slag/fly ash mixtures led to a slower setting due to the lower reactivity of the ash. The content of reactive CaO, the main chemical component of slag, decreased with the higher addition of the fly ash, leading to a lower amount of dissolved Ca^2+^ ions in the medium and a deceleration of the activation reaction [19,43,44]. Thus, the decelerated formation of C–A–S–H gel may lengthen the setting time of alkali-activated slag/fly ash pastes. The setting time of 100% BFS was directly related to the formation of calcium silicate hydrate (C–S–H) gel, but when the ash content was increased in the 85% BFS and 70% BFS samples, C–S–H gel also with sodium aluminosilicate hydrate gel (N–A–S–H) was formed [15].

In addition, Figure 1 shows the effect of different admixtures, vinyl copolymer (V), melamine (M) and three polycarboxylates (PC1, PC2 and PC3), on the initial and final setting time of the 100%, 85% and 70% BFS samples activated with sodium silicate. In the case of 100% BFS pastes, the presence of admixtures affected the setting times differently, PC1 accelerated the initial setting time by nearly 80 min, while PC2 had the opposite effect: retarding it by 50 min. However, the final sets were longer in all cases. In the case of 85% BFS pastes, the admixtures based on melamine and the three polycarboxylates retarded the initial setting, especially PC1. For these two admixtures, M and PC1, the final setting time was prolonged by 180 and 230 min, respectively. In the case of the 70% BFS pastes, the admixtures had no significant effect on the initial setting time, but PC1 lengthened the final set by 400 min, and, for its part, PC2 accelerated it by 190 min.

#### 3.1.2. Mini-Slump

The alkaline activation of the three slag/fly ash mixtures with sodium metasilicate showed good workability at the first moment (0 min) as is reported in Figure 2. An increase in the replacement of fly ash implied higher slump values. The smooth spherical shape of the ash was a determining factor in this improvement of workability [32], which was studied at 30, 60, 90, and 120 min. A decrease in slump values over time was observed in the three formulations. For the 100% BFS mixture, the slump value was reduced around 40% at 120 min, while these reductions were lower than 20% for 85% BFS and 70% BFS. Therefore, the rate of decrease in a slump over time decreased with the higher amount of fly ash due to the slower setting of fly ash compared to slag [45].

In Figure 2 results of the mini-slump tests of the alkali-activated slag/fly ash pastes with admixtures are also shown. In the 100% BFS sample, a clear improvement in the initial flow diameter was observed for V and PC3 admixtures. However, important retention of the slump between 0 and 120 min took place for V, indicating a possible chemical change in the alkaline solution, while PC3 presented flow loss during the first 30 min and then these values were maintained. On the other hand, M showed a slump value similar to that of the pastes without admixture in the first moments and retained its flowability for a longer duration, achieving the highest slump value. In the presence of admixtures, the flow of the 85% BFS paste improved at beginning of the test, except for V that was comparable to the slump for the control paste. The admixture based on melamine and PC3 improved the workability of the three alkali-activated mixes more efficiently during the full 120 min that the test lasted. According to the results, in the 70% BFS pastes, an improvement in the initial flow diameter was observed for all admixtures except for PC2, and a small reduction in the rate of flow loss with time was observed with the incorporation of all of them. PC3 displayed the highest rise in flowability.

The nature of the admixture played a significant role in controlling the workability of the mixes. While the vinyl copolymer had no impact on paste slump and setting time, melamine-based and polycarboxylate (PC3) admixtures increased the workability of fresh pastes in this high alkaline media and even retarded the setting. The addition of melamine-based admixtures, which contain the sulphonate group, implied adhesion to the slag or ash grains, giving the grain a negative electrical charge. The different grains were repelled by electrostatic repulsion, contributing to the release of the liquid component trapped in the flocs. It favoured the mobility of particles, improved the workability, and delayed the setting. In the case of PC3, the electrostatic repulsion was due to the absorption by the carboxyl group, but the presence of long and voluminous chains due to ethylene oxide (–CH_2_CH_2_O–) also allowed the separation of the grains by steric repulsion. This kept the particles sterically at distance, enhancing the mixture workability.

These variations in the effects of the admixture on the setting time and slump could be attributed to their different stability behaviours in the high-alkali media [28,30,46].

#### 3.1.3. Stability of Admixtures

The stability of the five admixtures (V, M, PC1, PC2 and PC3) in a sodium silicate medium for 24 h was studied by FTIR. Their infrared spectra are given in Figure 3. The absorption peaks at around 3410 and 1640 cm^−1^ in all the spectra were associated with the stretching and deformation vibration of the water molecules.

The FTIR spectra show that the chemical structure of all admixtures was modified when they were kept in a sodium silicate solution except for the melamine-based. The characteristic vibration bands of the vinyl copolymer (V) admixture appeared at 3438, 1600, and 702 cm^−1^ corresponding to the vibration of the N–H bond of the amine, at 1722 cm^−1^ corresponding to the vibration of the C=O bond of the carboxylic acid, and at 1183 and 1040 cm^−1^ corresponding to the vibration of the SO_3_^−^ groups [27]. When V was kept in the alkaline solution, the band at 1722 cm^−1^ disappeared, while two new bands were observed at 1568 and 1411 cm^−1^ that were associated with the vibration of carboxylate groups (COO^–^). This resulted from the alkaline hydrolysis of the amide that formed a part of this admixture to give rise to an amine that contained the sulphonate group and the carboxylate salt. These structural changes in the vinyl copolymer in the highly alkaline solutions (pH > 13) were previously observed by Palacios et al. [27], which could explain its lower superplasticising effect in the three formulations.

The FTIR spectrum of melamine (M) derivate showed four characteristics bands caused by the vibrations of the N–H bond: in the amine (1600 cm^−1^); the C–H bond (1470 cm^−1^) in the CH_2_ adjacent to a heteroatom; and the S–O bond (1190 and 1030 cm^−1^) in the sulphonate groups [30]. When this admixture was kept in the sodium silicate solution, its chemical formulations showed no structural alteration, which justified their good performance as superplasticizers: see Figure 2.

Finally, the adsorption bands at 1730, 1467, and 1350 cm^−1^ in the polycarboxylate (PC1, PC2 and PC3) admixtures belonging to the vibrations of the C=O, =CH_2_ and –CH_3_ groups, respectively [28,30,47,48]. In the sodium silicate solution, PC1, PC2 and PC3 admixtures underwent structural changes, which was deduced from the absence of a band at 1730 cm^−1^ and the detection of two bands around 1574 and 1410 cm^−1^ associated with the carboxylate groups (COO^–^). These alterations in their structures were due to the alkaline hydrolysis of these admixtures in highly alkaline media, where the ester groups gave rise to the respective ethers and carboxylate salts [27]. In PC1 and PC2, the steric hindrance from these ether chains was practically non-existent. The main chain with the carboxylate groups was adsorbed onto the surface of the slag and fly ash particles, while the lateral chains with ethers broke away from the main chain. However, PC3 had a different, non-comb, structure that presented long chains, and their steric effect may explain why such an admixture worked more efficiently relative to PC1 and PC2.

### 3.2. Effect of M and PC3 on the Reaction Kinetic and Hardened Properties of Alkali-Activated Slag/Fly Ash Mixtures

In light of the results, the melamine-based and polycarboxylate with long-chain admixtures resulted in the best dispersive and fluidifying properties over time for the three formulations. For this reason, a study using isothermal calorimetry was carried out to evaluate the kinetics of the reaction of the 100% BFS, 85% BFS and 70% BFS pastes with M and PC3. The mechanical strengths were also determined at 2 and 28 days.

#### 3.2.1. Isothermal Conduction Calorimetry

The rate of heat evolution and cumulative heat evolution of alkali-activated slag/fly ash pastes in the absence and presence of M and PC3 admixtures within the first 5 days is given in Figure 4. The heat release curves (Figure 4a) exhibited a pre-induction period (first peak) during the first hour of reaction. It corresponded to the wetting and dissolution of the raw materials, especially slag, and also partly due to the coagulation of the resulting dissolved silicate and aluminate units and their interactions with Ca and Na [36,43,49]. The peak was followed by an induction period with a low heat flow and reduced reaction. The new reaction products grew rapidly, and the surface of anhydrous slag particles was covered by a layer of these products. Therefore, the amount of available alkalis for slag dissolution was limited and the heat flow was reduced [50]. In this study, its duration was strongly dependent on both the binder composition (the nature and the reactivity of the raw materials) and the nature of the admixture. After this period, an intense peak (constituted by acceleration and deceleration periods) was observed and associated with nucleation, growth and precipitation of reaction products, C–A–S–H and N–A–S–H gels. The formation of these gels was the result of a condensation reaction between the silicate and the aluminate species and silicate present in the alkaline activator [36,43].

After the first heat-evolution peak, all mixes (100% BFS, 85% BFS and 70% BFS) exhibited an induction period that finished about 4, 6, and 8 h before the acceleration/deceleration peak, respectively. The elongation of the induction time was associated with the low reactivity of the fly ash. It can act as a nucleation site for the reaction products because it has a negative contribution to the further reaction between slag grains and alkalis. In addition, a long time was required to reach a critical ionic species concentration in solution to form reaction products.

The acceleration/deceleration peak appeared at around 9.3 h for the sample with 100% of the slag, this peak shifted to longer times with broader and lower intensities as the ash amount increased. When the samples contained 15% or 30% of fly ash, the acceleration peak was delayed at 13 h and 15.7 h and the peak height decreased by nearly 24% and 40% respectively. These differences in the second peak were mainly associated with the reduction of total slag content resulting in a smaller heat release rate and moderate reaction of ash at ambient temperature.

The calorimetric curves also showed that the presence of admixtures retarded the acceleration/deceleration peak and decreased its intensity. In the 100% BFS sample, the longest delay was induced by PC3, but in the other samples, 85% and 70% of slag, the signal had identical position and intensity. M and PC3 admixtures were adsorbed onto the slag and fly ash grains and exerted their superplasticiser function. Therefore, the presence of an admixture and a greater amount of fly ash retarded the initial alkaline activation. However, the nature of the admixtures did not seem to have a significant effect on the reaction kinetic at least in 85% and 70% BFS formulations.

The shape of the calorimetric curves in the presence of admixtures was characteristic of alkali-activated slag/fly ash cements, with the main signal associated with the precipitation of the reaction products [31].

The cumulative heat release of the alkali-activated slag/fly ash pastes within the first 120 h of reaction is shown in Figure 4b. The two rising portions depicted the heat release contribution due to the wetting and dissolution of slag and the acceleration phase of the reaction, respectively [51]. The relatively flatter region between these two rises corresponds to the induction period.

The cumulative heats were 113 J/g, 109 J/g and 103 J/g for the 100% BFS, 85% BFS and 70% BFS samples respectively at 5 days. Samples with a higher slag percentage exhibited higher total heat of reaction, indicating again that ash presented lower reactivity than slag under ambient temperature and that the alkaline activation occurred later.

In the first moments, the inclusion of M and PC3 decreased the activation heat in the three formulations. Their presence dispersed the flocs, retarding the precipitation of the reaction products and decreasing the heat release. However, eventually, around 48 h, the cumulative heat raised became quite similar to that obtained for their respective reference samples. The nature of the admixtures only had a slight influence on the total heats in the 70% BFS sample, where their values were 99 J/g and 105 J/g for M and PC3, respectively, at 5 days.

#### 3.2.2. Mechanical Strengths

Mechanical strengths for the three formulations with M and PC3 admixtures at 2 and 28 days are reported in Figure 5. The flexural strengths at 2 and 28 days of the three formulations were in the range of 4.9–6.8 MPa and 8.9–10.6 MPa respectively, but these values decreased with the replacement of slag, see Figure 5a. This same trend is observed in Figure 5b for the compressive strengths, but the differences between the three alkali-activated slag/fly ash mortars were more marked at both ages. As the slag percentage in the formulation decreased, a lower calcium content was provided to the medium and lower precipitation of the C–A–S–H gel took place. In addition, the reaction between Ca^2+^ dissolving from slag grains and silicate anions from the alkaline activators also played an important role in the setting and calorimetry in these systems, as previously observed in Figure 1 and Figure 4. A driving force was provided by the formation of the gel to accelerate the chemical reaction leading to larger strength of alkali-activated binders cured at room temperature [5,45].

As can also generally be seen from Figure 5a, the addition of M and PC3 led to small modifications in flexural strengths at 2 and 28 days independent of the composition of the alkali-activated slag/fly ashes formulations and even slightly increased these values, except for the 70% BFS sample, where the presence of PC3 attained a lower strength. Therefore, a higher percentage of fly ash and an admixture with high long chains exerting steric repulsion led to a clear loss of mechanical properties.

On the other hand, the effect of the admixtures on compressive strength is shown in Figure 5b. At 2 days, two different behaviours were observed according to the nature of the admixture. For PC3, the compressive values of the three formulations decreased, while for M they were maintained or increased. This behaviour was attributed to the good dispersion of the slag and/or fly ash particles with an improvement in the mixture workability, thereby obtaining dense and homogeneous mortars [29].

However, both types of admixtures negatively affected the compressive strength of the three samples about the control specimens without admixture in the long term, 28 days. In the 100% BFS samples, the drops in compressive strength were 12% and 14% for M and PC3, respectively; in the 85% BFS samples, they were 10% and 15%; and finally, in the 70% BFS samples, the drops were 9% and 16%. These reductions were related to an increase of the entrained or entrapped air in the mortar, as observed previously by [31], or the stability behaviour of admixtures in highly alkali media [30]. The loss of the compressive strength was somewhat greater in mortars with the polycarboxylate admixture.

#### 3.2.3. Porosity

Figure 6 shows the pore size distribution and total porosity measured by MIP method at 28 days. The total porosities of the 100% BFS, 85% BFS and 70% BPS mortars were 12.6, 9.4 and 11.3% respectively. The three samples exhibited the same two most-probable pores with a size of around 7–8 nm and 1–2.5 µm. The incorporation of fly ash in the formulations induced a reduction in total porosity. The fly ash may be acting as a filler due to the low reactivity at room temperature [52], refining the pore structure of the 85% BFS and 70% BFS samples. However, there was no clear trend between the values. Lee et al. [19] also observed that, as the amount of slag increased, the total porosity increased, since the pore volume in the region of the mesopores (2–50 nm) was greater, as can be seen in Figure 6.

Two different behaviours were observed in the pore size distribution and total porosity of the three alkali-activated slag/fly ash mortars with M and PC3. The total porosity decreased for the 100% BFS mortars in the presence of both admixtures, with a lower percentage of gel pores (size < 0.01 µm). Their presence improved the mobility of the particles, leading to better packing. For the 85% and 70% BFS mortars, the entrained air increased and the total porosity increased although it was higher for those with M. However, a refinement in porosity was detected with a rise in porosity of a smaller size. Therefore, the nature and dosage of the raw materials that constituted the alkali-activated cement seemed to condition how the additives affect porosity.

In light of the results, M and PC3 improved the workability of the three alkali-activated slag/fly ash blends for 2 h, even the presence of M admixture increased the early mechanical strengths due to their superplasticizer function. However, their impact on the hardened properties at 28 days was negative, thus reducing the compressive strengths and increasing the total porosity.

## 4. Conclusions

This study investigated the fresh and hardened properties of three alkali-activated slag/fly ash mixtures (100/0, 85/15 and 70/30) with five superplasticisers (vinyl copolymer, melamine and three polycarboxylates). The following conclusive statements hold:

A higher proportion of fly ash in the alkali-activated pastes increased workability over time and retarded setting due to its smooth spherical shape and low reactivity at room temperature. In addition, a retarded reaction kinetic and a decrease in mechanical strengths were observed because a lower calcium content was provided to the medium and a decelerated formation of C–S–H gel took place. Total porosities decreased with the incorporation of fly ash, so it may have acted as a filler.

The presence of melamine and PC3 polycarboxylate admixtures in the three alkali-activated slag/fly ashes pastes provided greater fluidity. Vinyl copolymer and polycarboxylates admixtures underwent alkaline hydrolysis in this sodium silicate medium. However, PC3 admixture exerted its plasticising function more effectively due to the presence of very long chains in its structure.

The presence of M and PC3 retarded the initial alkaline activation because they were absorbed onto the slag and ash grains thereby improving the workability of the three pastes. At 2 days, the compressive strengths were maintained or increased in the presence of M and PC3 admixtures, so that a good dispersion of these grains led to dense and homogenous mortars. However, these values decreased at long curing times due to the entrained or entrapped air in the mortar, which was also reflected in an increase in the total porosity of the 85% BFS and 70% BFS mortars.

## Figures and Tables

**Figure 1 materials-15-00992-f001:**
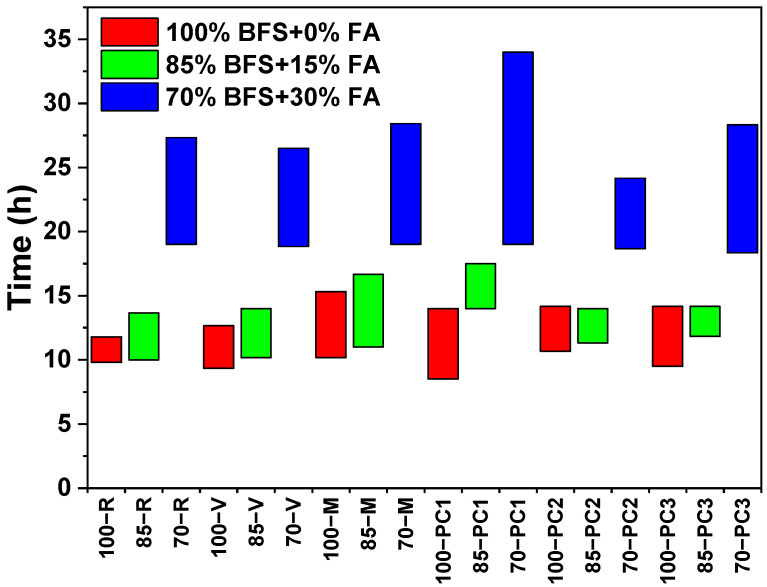
Setting time of different alkali-activated slag/fly ash pastes as a function of BFS/FA ratio and admixture.

**Figure 2 materials-15-00992-f002:**
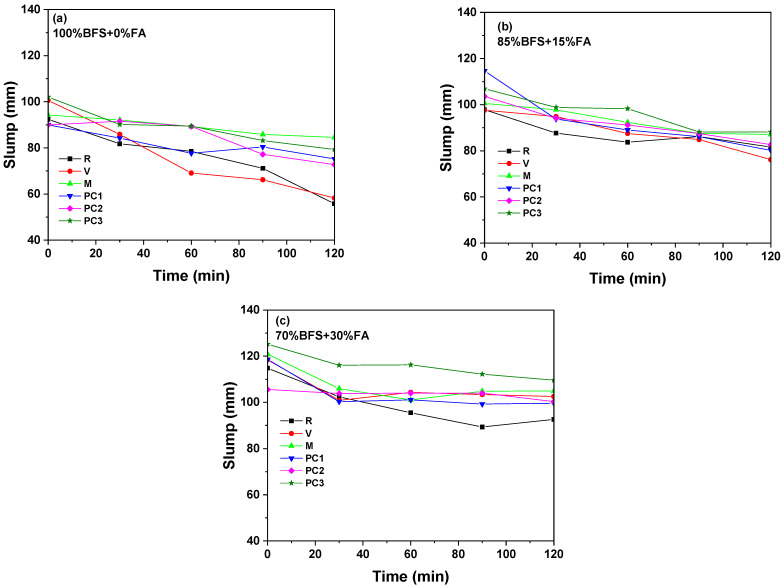
Evolution of the slump values with time for different binder proportions and admixtures. (**a**) 100% BFS + 0% FA, (**b**) 85% BFS + 15% FA, and (**c**) 70% BFS + 30% FA.

**Figure 3 materials-15-00992-f003:**
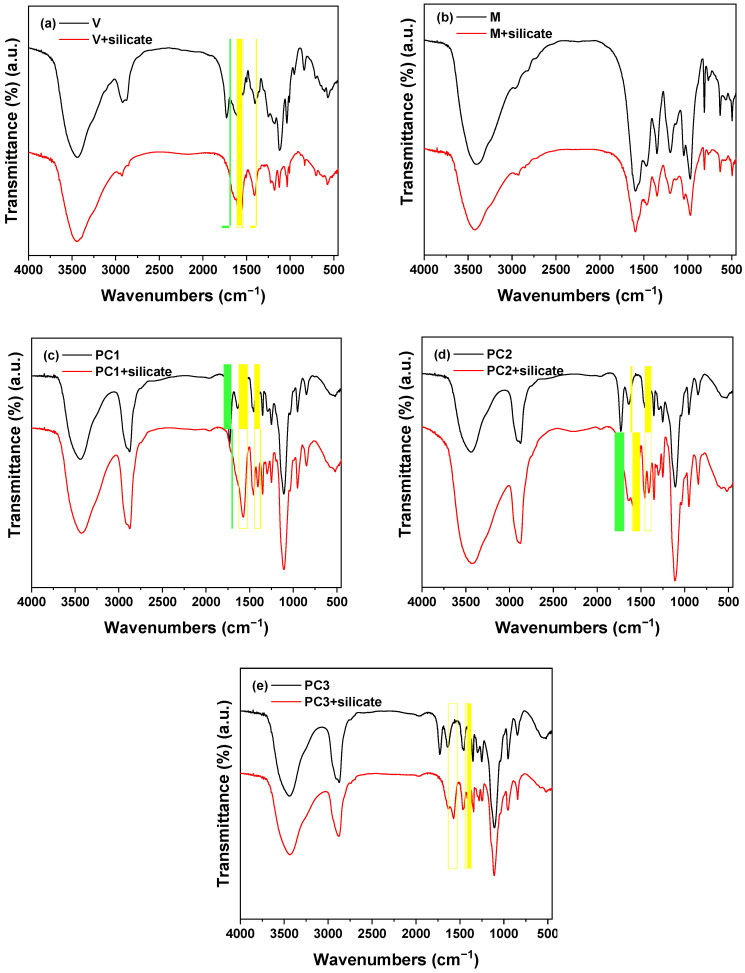
Infrared spectra of the five admixtures in the original state and sodium silicate medium: (**a**) V, (**b**) M, (**c**) PC1, (**d**) PC2, and (**e**) PC3. Green: a missing band; yellow: a new band.

**Figure 4 materials-15-00992-f004:**
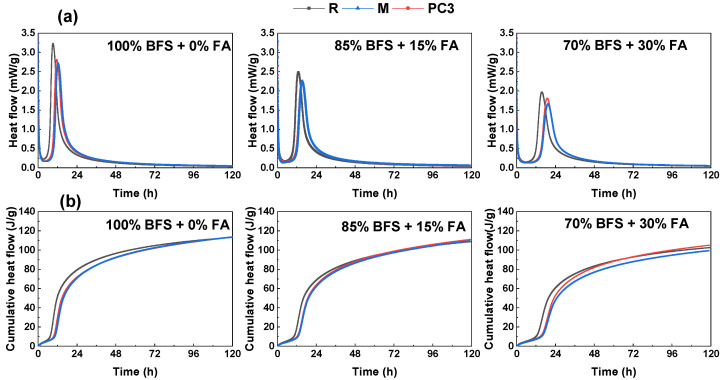
(**a**) Heat flow and (**b**) cumulative heat flow of the alkali-activated slag/fly ash pastes in the absence and presence of M and PC3 admixtures within the first 5 days.

**Figure 5 materials-15-00992-f005:**
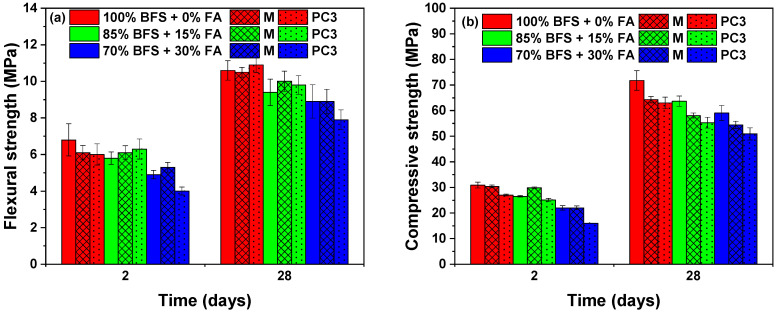
(**a**) Flexural and (**b**) compressive strengths of different binder proportions in the absence and presence of M and PC3 admixtures at 2 and 28 days.

**Figure 6 materials-15-00992-f006:**
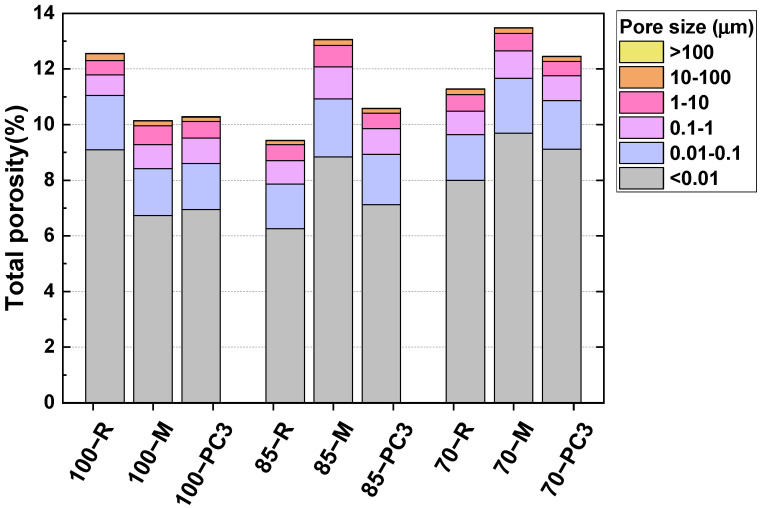
Pore size distribution and total porosity of alkali-activated slag/fly ash mortars in the absence and presence of M and PC3 admixtures at 28 days.

**Table 1 materials-15-00992-t001:** Chemical composition of BFS and FA used in this study.

Oxides (wt.%)	CaO	SiO_2_	Al_2_O_3_	MgO	SO_3_	TiO_2_	Fe_2_O_3_	K_2_O	Na_2_O	Others	L.O.I *
BFS	45.7	32.3	9.6	7.1	1.6	0.9	0.5	0.5	0.3	0.5	0.95
FA	4.8	42.4	27.0	0.8	1.4	1.1	18.4	1.5	0.5	0.5	1.60

* L.O.I am the loss on ignition at 1000 °C.

**Table 2 materials-15-00992-t002:** Characteristics and dosage of the five chemical admixtures.

Admixtures	V	M	PC1	PC2	PC3
Structure	Sulphonate groups and ester	Sulphonate groups	Comb-like structure (backbone and short side chain length)	Comb-like structure (backbone and long side chain length)	Structure with very long chain length
Action mechanism	Mainly electrostatic repulsion	Electrostatic repulsion	Mainly steric hindrance	Mainly steric hindrance	Mainly steric hindrance
Dosage (% binder)	1.00	1.20	1.00	1.25	1.10

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
