# Peer review of "Influence of the Fly Ash Content on the Fresh and Hardened Properties of Alkali-Activated Slag Pastes with Admixtures"

_materials, 2022, doi:10.3390/ma15030992_

Round 1
Reviewer 1 Report
In the results and discussion section, you must explain why:
-The presence of M and PC3 admixtures and a greater amount of fly ash re-tarded the reaction kinetic of all the alkali-activated slag/fly ashes pastes.
- At 2 days, the compressive strengths were maintained or increased in the presence of M and PC3 admixtures in the three mortars, but these values decreased at long curing times.
- It was not observed a clear trend in the total porosity of the 100%BFS, 85%BFS and 70%BFS mortars
Author Response
Reviewer 1
In the results and discussion section, you must explain why:
-The presence of M and PC3 admixtures and a greater amount of fly ash re-tarded the reaction kinetic of all the alkali-activated slag/fly ashes pastes.
- At 2 days, the compressive strengths were maintained or increased in the presence of M and PC3 admixtures in the three mortars, but these values decreased at long curing times.
- It was not observed a clear trend in the total porosity of the 100%BFS, 85%BFS and 70%BFS mortars.
The authors appreciate the review that has indicated that these aspects are not sufficiently explained so we have extended those parts clarifying the text. Results and discussion section has been also modified, but, as the referred sentences are taken from the conclusions, we have focus to expand the explanations in that section to include cause-effect aspects:
A higher proportion of fly ash in the alkali-activated pastes increased workability over time and retard setting due to its smooth spherical shape and low reactivity at room temperature. In addition, a retarded reaction kinetic and a decrease of the mechanical strengths were observed because a lower calcium content was provided to the medium and a decelerated formation of C-S-H gel took place. Total porosities decreased with the incorporation of fly ash so that it may be acted as a filler.
[…]
The presence of M and PC3 retarded the initial alkaline activation because they were absorbed onto the slag and ash grains improving the workability of the three pastes.
At 2 days, the compressive strengths were maintained or increased in the presence of M and PC3 admixtures, so that the good dispersion of these grains leads to dense and homogenous mortars. However, these values decreased at long curing times due to the entrained air or entrapment air content in the mortar, which was also reflected in an increase in the total porosity of the 85%BFS and 70%BFS mortars.
Reviewer 2 Report
This study investigated the fresh and hardened properties of three alkali-activated slag/fly ash mixtures (100/0, 85/15 and 70/30) with five superplasticisers (vinyl copolymer, melamine and three polycarboxylates). Below are some points to be inquired before publishing:
- On what basis were the proportions of fly ash replaced by iron slag determined?
- Authors are asked to provide standard specification for slump procedure
- Studying the characteristics at two ages is not sufficient to judge the behavior of the designed alkali-activated cement-based geopolymer
- On what basis was sodium metasilicate chosen as an activator and why was it determined as 7% of the binder?
- The conclusion section needs further revision to make it easier for the reader
Author Response
First of all, the authors want to thank the reviews and comments made in the revision, helping the paper to be more clear and enriched. Please, find the answer to each point below:
- On what basis were the proportions of fly ash replaced by iron slag determined?
To answer this question please allow us to contextualize this work. This study is carried out within the framework of the IRINEMA project (Immobilization of Nuclear grade Ion Resins in Alkali-activated Materials). The aim of it is to find an alkaline activated material to replace the Portland-based cement currently used to immobilize nuclear waste. Slag and ash proportions are chosen to take as a starting point the formulation currently used in Spain for this purpose, which is approximately 70% OPC - 30% FA. It was decided to substitute the OPC for BFS to carry out comparative studies (70% BFS) and we also want to evaluate the behaviours of a total substitution by slag (100% BFS) and an intermediate value to evaluate possible trends (85% BFS).
- Authors are asked to provide standard specification for slump procedure.
Although mini-slump test is a widely used cement paste evaluation method due to its advantages, there is no official standardization of the procedure. Authors have followed the procedural guidelines proposed in:
Tan Z, Bernal SA, Provis JL. Reproducible mini-slump test procedure for measuring the yield stress of cementitious pastes. Mater Struct. 2017;50(6):235. doi: 10.1617/s11527-017-1103-x. Epub 2017 Oct 19. PMID: 31997916; PMCID: PMC6956902.
This reference used for slump procedure has been included in section 2.3 Testing procedure (ref [37]).
- 3. Studying the characteristics at two ages is not sufficient to judge the behavior of the designed alkali-activated cement-based geopolymer.
These two curing ages have been chosen to see the effect of the admixtures in the fresh state, for that reason they have not been done so far at older ages. Indeed, authors agree with the reviewer that if the geopolymer design wanted to be validated, it would be necessary to carry out more in-depth chemical and physical characterization considering other studies and at different curing ages, but we think that those can be the subject of a publication in itself, and would be outside the scope of this paper.
- On what basis was sodium metasilicate chosen as an activator and why was it determined as 7% of the binder?
Sodium metasilicate was chosen as the activator because it is one of the most used activators in alkaline activation. Thanks to its high pH (13.5 in this study) and the nature of the reaction products it forms, it has high mechanical performance and durability, as reported in previous studies:
- Provis, J. L. (2009). Activating solution chemistry for geopolymers. In Geopolymers (pp. 50-71). Woodhead Publishing.
- Provis JL, van Deventer JSJ, eds. 2014. Alkali-Activated Materials: State-of-the-Art Report, RILEM TC 224-AAM. Dordrecht, Neth.: RILEM/Springer
- Provis, J. L., & Bernal, S. A. (2014). Geopolymers and related alkali-activated materials. Annual Review of Materials Research, 44, 299-327.
On the other hand, sodium metasilicate is used in solid state because in industrial applications, the silicate can be produced on-site and transportation problems, such as possible spills, are avoided.
The percentage has been set at 7% because in a previous study it has been seen that this was the dose that achieved the greatest reactivity.
Criado, M.; Walkley, B.; Ke, X.; Provis, J.L.; Bernal, S.A. Slag and activator chemistry control the reaction kinetics of sodium metasilicate-activated slag cements. Sustain. 2018, 10, doi:10.3390/su10124709.
These explanations and references have been included in the manuscript in section 2.1 Materials.
- The conclusion section needs further revision to make it easier for the reader.
To make it easier for the reader authors have reviewed and expanded the explanations in the conclusions section including cause-effect aspects.
Reviewer 3 Report
Please check the attached file.

Author Response
We appreciate all the reviews and comments of Reviewer 3. The authors have tried to include each of them in the manuscript and corrected the formatting errors. The references requested in each case have been included as well as a paragraph at the end of section 1. Introduction clarifying the novelty and objective of the study. Besides, the authors have reviewed and expanded the explanations in the conclusions section including cause-effect aspects to improve this section.
Regarding the figures, authors were not sure what they were specifically asked to do as “rearrange the figures”, that is why they have been placed in a different way hoping that this was the comment, if it is not, please let us know so that we can modify it.
Also, the authors want to apologize for not including photos of the raw materials and curing and testing conditions in the manuscript. The raw materials, the procedure to carry out the tests, and the tests are the common and traditional ones employed in the construction field. For this reason, authors consider that they are not particularly interesting and prefer not to include them in the manuscript.
Reviewer 4 Report
- What is the novelty of this article? Slag, fly ash and superplasticizers (vinyl copolymer, melamine, and three polycarboxylates) have been used in alkali-activated pastes in some studies.
- What is the major difference in the sixth and seventh paragraphes in introduction part?
- There are some writing errors, for example Na2SiO3 should be changed to Na2SiO3, and cm3 should be changed to cm3.
- To make the discussion more clear, section 3.1 is suggested to be divided into several parts labelled as 3.1.1, 3.1.2,etc.
- For the same reason, section 3.2 is suggested to be divided into several parts also.
Author Response
First of all, the authors would like to thank all the comments and revisions made by Reviewer 4, since we believe that, after these revisions, the article has been enriched and improved. We, therefore, proceed to comment on each of the reviews:
- What is the novelty of this article? Slag, fly ash and superplasticizers (vinyl copolymer, melamine, and three polycarboxylates) have been used in alkali-activated pastes in some studies.
The novelty of this work is to study the plasticizing properties of new commercial admixtures that could solve problems that have been observed when admixtures interact with strongly alkaline media. Although all the admixtures are of recent design and marketing, it can be worth highlighting the vinyl copolymer (V), which, apart from sulfonates as it has traditionally included, also has ester groups, and additives based on polycarboxylates that have different chain lengths. Furthermore, the combination of slag and fly ash in different percentages with additives has not been enough studied.
To clarify this aspect based on the review, a paragraph has been included at the end of section 1. Introduction.
- What is the major difference in the sixth and seventh paragraphes in introduction part?
The difference is that in the sixth paragraph the results are from studies in which the percentages of ash and slag have been kept fixed and, in the seventh, the percentages of both raw materials have been varied.
The authors have reviewed that part of the introduction to clarify this point.
- There are some writing errors, for example Na2SiO3 should be changed to Na2SiO3, and cm3 should be changed to cm3.
Writing and formatting errors have been corrected throughout the manuscript.
- To make the discussion more clear, section 3.1 is suggested to be divided into several parts labelled as 3.1.1, 3.1.2,etc
Authors agree with reviewer 4 so section 3.1 Influence of the fly ash content on the fresh properties of alkali-activated pastes with admixtures has been divided into sections 3.1.1. Setting time, 3.1.2. Mini-slump and 3.1.3. Stability of admixtures, with the aim of facilitating reading and clarifying the discussion.
- For the same reason, section 3.2 is suggested to be divided into several parts also.
Authors agree with the suggestion of reviewer 4 so section 3.2 Effect of M and PC3 on the reaction kinetic and hardened properties of alkali-activated slag/fly ash mixtures has been divided into sections 3.2.1 Isothermal conduction calorimetry, 3.2.2. Mechanical strengths and 3.2.3. Porosity.
Reviewer 5 Report
Some revisions are important to improve the quality of this work before it can be published.
1. The abstract section should focus more on the experimental methed and test results, the existing one must be revised to be brief.
2. The languge in this work needs to be improved, the sentences are readable but sometimes confusing.
3. The quality of Fig. 3 needs to be improved, its style is different from other figures in this work.
4. The analysis of the pore structure based on MIP results is too simple. Some studies can be referenced to improve the analysis in this section:
1) Multi-structural evolution of conductive reactive powder concrete manufactured by enhanced ohmic heating curing
2) Effect of carbonation curing on sulfate resistance of cement-coal gangue paste
3) Ohmic heating curing of high content flfly ash blended cement-based composites towards sustainable green construction materials used in severe cold region
5. This work investigates the fresh and hardened properties of the samples, however the microstructural analyses are limited, please explain the reason.
Author Response
The authors would like to acknowledge the comments and reviews of reviewer 5 that help improve the quality of this paper. Please find the answers to each point below:
- The abstract section should focus more on the experimental methed and test results, the existing one must be revised to be brief.
The abstract section has been revised and shortened in the manuscript.
- The languge in this work needs to be improved, the sentences are readable but sometimes confusing.
The authors have checked English stylistics for clarifying the text.
- The quality of Fig. 3 needs to be improved, its style is different from other figures in this work.
The authors were not sure what was specifically required to modify in Figure 3 so we have rearranged the figure and added the title of the vertical axis in each graph. If those changes are not enough, please let us know so we can fix them.
- The analysis of the pore structure based on MIP results is too simple. Some studies can be referenced to improve the analysis in this section:
1) Multi-structural evolution of conductive reactive powder concrete manufactured by enhanced ohmic heating curing
2) Effect of carbonation curing on sulfate resistance of cement-coal gangue paste
3) Ohmic heating curing of high content flfly ash blended cement-based composites towards sustainable green construction materials used in severe cold region
Thank you very much for your comment, the pore structure analysis has been improved and a reference has been included to support the results. The authors appreciate the bibliography provided by reviewer 5 and will take it into account in future experimental planning. Since it cannot apply to this study because it is performed on alkali-activated materials and the results and discussion differ.
- This work investigates the fresh and hardened properties of the samples, however the microstructural analyses are limited, please explain the reason.
Authors consider that with the tests proposed in this work the study of the properties in fresh and hardened states is addressed, and we believe that a mineralogical and microstructural study would be very interesting as a next step but also that deep study may constitute another publication in itself.
Round 2
Reviewer 3 Report
The authors have addressed the comments and I have no additional comments.
Author Response
The authors appreciate the comments and revisions of reviewer 3.
Reviewer 4 Report
The article has been revised according to my comments, however I still think that the novelty is not enough to make the article be accepted.
Author Response
The authors have included an additional paragraph in the introduction framing the project and clarifying its novelty.